# Real-Time Detection of Non-Stationary Objects Using Intensity Data in Automotive LiDAR SLAM

**DOI:** 10.3390/s21206781

**Published:** 2021-10-13

**Authors:** Tomasz Nowak, Krzysztof Ćwian, Piotr Skrzypczyński

**Affiliations:** Institute of Robotics and Machine Intelligence, Poznan University of Technology, 60-965 Poznan, Poland; tomasz.nowak@doctorate.put.poznan.pl (T.N.); krzysztof.cwian@put.poznan.pl (K.Ć.)

**Keywords:** 3-D LiDAR, SLAM, intensity data, motion segmentation, deep learning

## Abstract

This article aims at demonstrating the feasibility of modern deep learning techniques for the real-time detection of non-stationary objects in point clouds obtained from 3-D light detecting and ranging (LiDAR) sensors. The motion segmentation task is considered in the application context of automotive Simultaneous Localization and Mapping (SLAM), where we often need to distinguish between the static parts of the environment with respect to which we localize the vehicle, and non-stationary objects that should not be included in the map for localization. Non-stationary objects do not provide repeatable readouts, because they can be in motion, like vehicles and pedestrians, or because they do not have a rigid, stable surface, like trees and lawns. The proposed approach exploits images synthesized from the received intensity data yielded by the modern LiDARs along with the usual range measurements. We demonstrate that non-stationary objects can be detected using neural network models trained with 2-D grayscale images in the supervised or unsupervised training process. This concept makes it possible to alleviate the lack of large datasets of 3-D laser scans with point-wise annotations for non-stationary objects. The point clouds are filtered using the corresponding intensity images with labeled pixels. Finally, we demonstrate that the detection of non-stationary objects using our approach improves the localization results and map consistency in a laser-based SLAM system.

## 1. Introduction

Although the Global Positioning System (GPS) is commonly used for outdoor localization there are still many scenarios in which autonomous vehicles, such as self-driving cars, have to localize themselves using their exteroceptive sensors exclusively. Such situations are typical in urban driving scenarios, due to tunnels, underground parking lots, and tall buildings. The sensors of choice for GPS-independent localization in autonomous vehicles are 3-D LiDARs [1,2], which provide reliable range measurements not affected by lighting conditions (also at night) and which are robust to weather changes.

Most of the LiDAR-based localization methods register consecutive observations using a variant of the well-known Iterative Closest Points (ICP) algorithm [3], assuming that the observed scene is static and rigid. These assumptions make these SLAM or LiDAR odometry solutions prone to errors caused by non-stationary objects present in the field of view (Figure 1, see also a short video (https://youtu.be/_wvu77xbUIQ, accessed on 16 August 2021)). Although there are SLAM algorithms that integrate Detection and Tracking of Moving Objects (DATMO) techniques [4], these systems are complicated because of the need to track each object separately, and typically can detect and handle only a limited number of such objects. This approach is useful whenever we need to keep track of the dynamics of some objects populating the scene, for example, for the purpose of maneuver planning, but has a high computational cost.

Hence, we introduce in this paper a category of non-stationary objects. Objects are considered as belonging to this category if they can move or are in motion, but also if their surface is not “rigid”, as a lawn area shaken by the wind, or tree canopies. The detection and segmentation of the LiDAR points belonging to such non-stationary objects are of particular importance in urban environments, where vehicles, pedestrians, and vegetation can be disregarded for localization, as there are enough man-made static surfaces [5]. A method that detects these objects in the context of automotive SLAM should process the incoming scans in real-time, using only the information that is yielded by the LiDAR itself. Such a method can be applied as a “filter” in the input of the LiDAR-based SLAM algorithm, disregarding the 3-D points that belong to non-stationary objects.

These requirements are met by the simple, yet effective learning-based approach to the segmentation of 3-D LiDAR data proposed in this paper. A distinctive feature and novelty in our approach is that it exploits the intensity output that is available in many 3-D laser scanners. While LiDAR intensity is used in SLAM [6] and place recognition to enhance the descriptiveness of LiDAR perception [7], this modality was so far not used for motion segmentation. Although our method uses already existing neural network architectures, we consider it an attempt to engineer a practical solution to a problem we have observed while developing PlaneLOAM [5], a feature-based LiDAR SLAM inspired by the LOAM (Lidar Odometry And Mapping) [8] processing pipeline.

The approach we describe in this article is based on a computation efficient algorithm for the synthesis of 2-D intensity images from 3-D LiDAR data, and is an extended version of our contributions presented at the local PP-RAI event [9], and then at the IEEE ICRA 2020 Workshop on Sensing, Estimating and Understanding the Dynamic World [10]. The journal article extends the description of the state-of-the-art and provides a more comprehensive evaluation of the method involving data from different LiDAR models and using the popular KITTI dataset [11]. The use of publicly available LiDAR sequences and the open source code of our system which is publicly available (https://github.com/kcwian/feature-based-LOAM, accessed on 16 August 2021) make it possible to replicate our results and use this approach in applications other than the PlaneLOAM system.

The practical value of our method pertains to its ability to segment out the LiDAR points belonging to non-stationary objects using only the LiDAR measurements and a neural network model, while the training process of this model involves a very limited human effort owing to the applied concepts of unsupervised learning and cross-modal transfer learning. The contribution of this work is threefold:a novel algorithm for synthesizing intensity images from 3-D LiDAR data;new methods for training of the neural network models in the detection of non-stationary objects, using commonly available grayscale images;a thorough evaluation of the detection and rejection of non-stationary objects using intensity images as the improvement of feature-based LiDAR SLAM.

The remainder of this article is organized as follows. In Section 2 we review a number of prior publications that are relevant to our work, while Section 3 and Section 4 describe in more detail the structure of the proposed method, and the neural network models with the training strategies, respectively. The integration of our motion segmentation method in the PlaneLOAM SLAM system is detailed in Section 5. Section 6 describes the experiments and their qualitative and quantitative results, while Section 7 concludes the paper.

## 2. Related Work

While a significant part of the research related to Simultaneous Localization and Mapping (SLAM) is devoted to passive visual perception [12,13], recently the interest in robust, real-time perception for autonomous cars has resulted in several technological innovations in the area of LiDARs. An example is the Ouster LiDAR that measures range, signal-intensity, and ambient light values, which are spatially and temporally aligned, without shutter effects [14]. Other new LiDARs that recently appeared on the market, such as new models of Velodyne sensors (e.g., Puck Hi-Res), are developed in response to the demand for accurate, active sensing in many application areas, and in autonomous driving in particular [1]. There are many tasks and environments that do not guarantee good lighting conditions and an abundance of visually salient features, while those conditions are a prerequisite for passive vision-based SLAM to work properly [15]. Moreover, LiDAR sensors provide reasonably dense depth images within the range of tens or even hundreds of meters and make it possible to build dense 3-D maps of the environment.

In this context, it is important to obtain information about the dynamic objects that are observed by a LiDAR on the scene. In automotive applications detecting such objects is essential to track the motion of other vehicles, and to ensure the safety of pedestrians and cyclists [16]. However, in this paper, we focus on LiDAR-based SLAM and visual odometry as the application domain that has its specific requirements concerning both the quality of range measurements and real-time processing [1,6].

The two main families of methods for detection of moving objects in LiDAR data that are reported in the literature are based on occupancy mapping or explicit tracking of selected objects. Algorithms in the former group, represented e.g., by [17,18], construct grid-based maps of the environment and exploit the fact, that the cells occupied by static structures (buildings, fencing, curbs, etc.) are updated more frequently than the cells pertaining to moving objects. Hence, the cells of high certainty (or probability, depending on the applied mathematical framework) of occupation represent the static part of the environment. Grid-based methods are easy to implement, but they do not scale well with the size of the map and require a number of re-observations of the same object to update all cells belonging to the object. Moreover, grid-based mapping algorithms use ray-tracing techniques to update the occupancy values, which may have prohibitive computation costs for modern multi-beam LiDARs. Object tracking algorithms can identify the objects of interest (e.g., cars) and compute their motion direction and velocity [19]. However, such algorithms typically use bounding boxes to segment the tracked objects from the whole point cloud, which results in an inaccurate classification of the LiDAR points located nearly the borders of such boxes and leaves misclassified points in the segmented point cloud. Such points may later cause problems in LiDAR-based SLAM because some invalid points remain in the map. If RGB-D sensors are employed, the visual information can be exploited to detect dynamic objects in the scene either in the feature-based [20] or direct [21] approach to SLAM.

Recently, Jo et al. [22] have proposed a real-time motion segmentation algorithm that divides LiDAR point clouds into static and dynamic parts employing the geometric relations between the consecutive scans yielded by a LiDAR and the properties of the sensor’s laser beam. Although this algorithm is computationally efficient, it requires accurate pose estimates of the moving LiDAR, which are obtained from an external source (IMU is used in [22]). This raises an issue if the integration into a SLAM system is considered, as our SLAM algorithm yields accurate pose estimates at a much lower frequency than the IMU sensor. Therefore, we are looking for a motion segmentation method that uses only the scans obtained directly from the LiDAR, to use it as a “filter” on the input of the SLAM system.

The enormous success of deep learning approaches in computer vision that proliferated also to 3-D LiDAR data processing [23,24], motivated research on the application of learning methods to visual odometry and SLAM [6]. However, the application of deep learning to LiDAR data turned out to be much more problematic than image processing, because of the lack of a representation of the acquired scans that is feasible for neural computations. Point clouds that are the most typical representation of 3-D range measurements are sparse, not differentiable, have a variable density of the measured points that are ordered at random in the data structure.

Therefore, the neural architectures proposed so far for the processing of point clouds ensure invariance to the point-order permutation, like PointNet [25], or employ a voxel representation [24]. Although a version of PointNet was applied to extract sparse features in the L^3^-Net LiDAR-based SLAM system [26], such architectures are suitable rather for small point clouds, and are considered too heavy with respect to the computation and memory burden, to be applied in real-time processing of LiDAR scans acquired in autonomous driving scenarios. The problem of promoting more trustworthy LiDAR readouts in SLAM was put forward in the recent deep learning LiDAR-based odometry system [27], which leverages the normal vectors’ consistency to compute a map of weights to the unsupervised loss function based on an ICP formulation. This confidence map contains higher weights for smooth surfaces and lower weights for trees or vehicles. However, the weighting mechanism is based on local surface properties and does not eliminate large moving objects such as trucks or trains.

Whereas [27] is an example demonstrating that weights for the LiDAR point clouds can be learned as a part of an end-to-end SLAM or visual odometry pipeline, a drawback of an entirely learnable SLAM system is that its performance may degrade when such a system is confronted with a previously unseen environment. Hence, in this paper, we focus on a relatively lightweight and computation efficient learnable filter for removing and/or weighting the range measurements depending on their assumed suitability for localization. The whole SLAM architecture we experiment with is built around a model-based approach [5], and we demonstrate that the accuracy of the estimated trajectories increases if we use our filter to eliminate the non-stationary objects.

Huge data streams produced by modern LiDARs: up to 2.2 million points per second from the Velodyne HDL-64E and about 2.6 million points per second from Ouster OS1-128, require a neural network model that avoids 3-D convolutions and complicated pre-processing operations. Such a representation is the range image, which can be seen as a cylindrical projection of the points obtained from a rotating multi-beam scanner. This concept was used in [28] with post-processing by Conditional Random Fields (CRF) to refine the results. More recently, Biasutti et al. [29] demonstrated that it is possible to adopt the proven U-Net semantic segmentation network to the 3-D LiDAR data by converting the point cloud into a 2-D image. However, both [28,29] used for training the bounding boxes of the KITTI object detection dataset [11], due to the lack of large outdoor datasets providing point-wise semantic information for LiDAR data. The KITTI object benchmark is used for training also by Dewan et al. [23]. This work is close to our approach with respect to both the LiDAR data representation as a 2-D image, and to the objective—segmentation of the LiDAR data into non-movable, movable, and dynamic points. However, the underlying semantics in [23] are quite different than ours, as they consider vegetation as static (non-movable) points, effectively reducing the task to binary classification, as cars are the only objects considered as movable. Dynamic objects are detected as a sub-class of the movable ones by estimating point-wise motion from two consecutive laser scans in post-processing.

As obtaining point-wise semantic annotations for 3-D point clouds is labor-intensive and time consuming, few datasets support this level of annotation. An exception is the recent SemanticKITTI [30], which was used for training of the RangeNet++ architecture [31] resulting in very accurate semantic segmentation, even on small objects. Nevertheless, building such datasets as SemanticKITTI requires a lot of effort, and methods that make it possible to train a neural network without explicit ground truth labels on LiDAR data are investigated. Similarly to our work, Piewak et al. [32] adopt the notion of cross-modal transfer learning [33], and use the class labels from RGB images in LiDAR data segmentation. However, their approach differs significantly from our concept, as they work with paired RGB-depth images produced by a custom sensory setup. The images are segmented using an established deep learning framework, and then the labels are projected onto the corresponding point clouds to produce the final training dataset. This approach, called autolabeling, keeps the manual annotation effort low, but requires the production of a custom dataset and is prone to calibration and time synchronization issues in the sensory setup [34].

## 3. The Approach to Detection of Non-Stationary Objects

### 3.1. Structure of the Solution and Training Strategies

The proposed methods are aimed at detecting points that belong to objects that do not provide a stable (i.e., stationary) reference for LiDAR-based localization and mapping. According to our assumptions, non-stationary objects are detected exclusively from the synthesized intensity images, using the learned semantics of the environment, and in the unsupervised version, also short sequences of the consecutive intensity images. We do not post-process point clouds to detect motion, we do not identify individual objects and do not track them. Although such an approach collects less information about the scene dynamics than the more typical DATMO algorithms (e.g., [35]), its advantage is the real-time performance and independence from any external information, such as an accurate pose estimate of the sensor.

Our key insight is that modern LiDARs yield range and signal-intensity, which are spatially and temporally aligned, while the spatial density of these measurements makes it possible to consider the intensity images as low-resolution camera images. This is particularly true when multi-beam LiDARs are used, which provide a relatively dense scan of the scene. The closer is the wavelength to the visible light spectrum, the more similar is the intensity output to a passive camera image. Using a shorter wavelength in the sensor it is possible to minimize the dependency of the intensity measurements on factors other than reflectivity [14]. Obviously, a LiDAR has a much lower spatial resolution (thus the density of the synthesized “pixels”), but we make an assumption that the visual differences that allow us to distinguish different objects in grayscale images are to a large extent preserved also in the LiDAR intensity data.

Our approach facilitates learning from standard grayscale images, which are commonly available with class and instance-level point-wise annotations. The approach is modular: the main part is the synthesis of grayscale-like images from the intensity LiDAR data and the training procedure that exploits images from commonly available datasets, which are modified for better compatibility with the synthesized ones.

An important feature of our approach is the possibility to apply it to different neural network architectures. We present results for two different networks: the ERFNet (Efficient Residual Factorized convNet) [36], a state-of-the-art supervised deep learning model for efficient semantic segmentation of images, and the Competitive Collaboration [37], a framework for unsupervised learning of optical flow and segmentation of a video sequence into the static scene and moving regions. Indeed, with the latter framework, we show that we even do not need annotated grayscale images—the network learns the segmentation task from the sequence of images in an unsupervised manner, and then transfers successfully to the LiDAR intensity domain using synthesized images. In both cases, the obtained segmentation is projected to the depth data domain and used to decide if particular 3-D points should be used or not by the SLAM system.

### 3.2. Synthesis of Lidar Intensity Images

Typical LiDARs used for outdoor navigation, such as Velodyne and Ouster sensors, measure distances at selected horizontal and vertical angles, returning also the corresponding intensity values. A single scan can be represented as an image, with each pixel mapped according to the angle increment of its corresponding laser beam, which results in a non-linear projection. An intensity image of 1024×64 [pix] resolution created this way from example data of the Ouster OS1-64 sensor is shown in Figure 2A.

However, to facilitate cross-modal transfer learning of neural networks we need to represent LiDAR measurements as an image compatible with the grayscale camera images, and produced according to the pin-hole camera model. To this end, we divide the 360° laser scan into six horizontal parts and synthesize six images with the assumed vertical field of view of ψ degrees. We define a virtual camera with the focal lengths $f_{x}$ [pix] and $f_{y}$ [pix] (anisotropic images), and the image center at $c_{x},c_{y}$ [pix]. Each pixel intensity of the *i*-th image ($u_{\mathrm{dest}}^{i},v_{\mathrm{dest}}^{i}$) is computed as the value for subpixel position ($u_{\mathrm{laser}},v_{\mathrm{laser}}$) from the raw LiDAR data: (1)$$\begin{array}{c} \theta =\arctan\left(u_{\mathrm{dest}}^{i}-c_{x},f_{x}\right)+\psi i,\;\;\;\phi =\arctan\left(c_{y}-v_{\mathrm{dest}}^{i}\cos\left(\theta \right),f_{y}\right), \end{array}$$(2)$$\begin{array}{c} u_{\mathrm{laser}}=\theta \frac{n_{\mathrm{hor}}}{{\mathrm{fov}}_{\mathrm{hor}}}+\frac{n_{\mathrm{hor}}+1}{2},\;\;\;v_{\mathrm{laser}}=\phi \frac{n_{\mathrm{ver}}}{{\mathrm{fov}}_{\mathrm{ver}}}+\frac{n_{\mathrm{ver}}+1}{2}. \end{array}$$

Finally, the image intensity for subpixel positions is computed using a linear interpolation of the neighboring pixel values, and an inpainting algorithm based on the Fast Marching Method [38] is applied to each image to remove no-return areas resulting from invalid measurements. Each of the synthesized images (examples are shown in Figure 2B–D) has the resolution of 1024×512 [pix].

The procedure described by (Equation 1) and (Equation 2) has to be parametrized to fit the particular LiDAR sensor model. The Ouster OS1-64 provides $n_{\mathrm{hor}}=1024$ horizontal measurements with the horizontal field of view ${\mathrm{fov}}_{\mathrm{hor}}=360$°, and $n_{\mathrm{ver}}=64$ vertical measurements with the vertical field of view ${\mathrm{fov}}_{\mathrm{ver}}=32$°. The remaining parameters for the OS1-64 images are: ψ = 60°, $f_{x}$ = $f_{y}$ = 800 [pix], $c_{x}=512$ [pix], $c_{y}=256$ [pix]. In the recent Ouster OS1-128 LiDAR, the geometry of measurements is the same, both sensors rotate at 10 Hz, but because of the higher resolution than in the OS1-64 model, the synthesized images have the vertical field of view of ψ = 90°, $n_{\mathrm{hor}}=2048$ measurements, ${\mathrm{fov}}_{\mathrm{ver}}=43$°, and $f_{x}=f_{y}=510$ [pix], while the remaining parameters are unchanged.

Intensity images synthesized from the Velodyne HDL-64E data (KITTI dataset) have different parameter values: $n_{\mathrm{hor}}=2083$ measurements, ${\mathrm{fov}}_{\mathrm{ver}}=22.3$°, $f_{x}=f_{y}=880$ [pix]. Figure 3 compares a synthesized HDL-64E intensity image (A) to its closest grayscale counterpart (B) in the KITTI dataset, selected on the basis of timestamps.

## 4. Learning Strategies for the Intensity-Based Motion Segmentation Task

The concept of learning how to detect non-stationary objects in the synthesized intensity images was implemented in two variants: supervised and unsupervised.

The supervised version exploits the idea of semantic segmentation for motion detection [23], assuming that the objects belonging to some predefined classes, such as cars, pedestrians, buildings, trees can be mapped to some other classes with respect to their dynamics. That is, we can assign to each object a label “stationary” or “non-stationary” relying purely on its semantics. An advantage of this approach is that we need only to perform semantic segmentation, which for camera images can be accomplished in real-time using a deep learning neural model [36]. A disadvantage is that with the purely semantic approach to detection we cannot handle well objects that can move but also can stand still. An obvious example is cars, which are the most numerous dynamic objects in urban SLAM scenarios, but they can be also parked alongside a road, and then can be used as static references by a SLAM system. Another disadvantage is that supervised learning requires large, labelled datasets, which are hardly available for the task we are aiming to. Therefore, we conceived a method that uses three semantic classes: stationary, possibly stationary, and non-stationary to embrace the diversity of possible object behaviors in the considered SLAM scenario. We also aimed at minimizing the effort required to implement the training process by using the cross-modal transfer learning concept. Therefore, we employ the recent, very efficient ERFNet architecture [36] (https://github.com/Eromera/erfnet, accessed on 16 August 2021) for real-time segmentation of images. The training process demonstrates the usefulness of grayscale images commonly available with pixel-wise annotations for cross-modal learning.

The unsupervised version, which is considered the final variant of our approach, is based on the lessons learned with the preliminary experiments with the variant based on ERFNet [9], and tries to overcome its limitations without much increase in the training effort. In the automotive SLAM application an important problem with the object detection method trained with camera images was the difference in the field of view between a typical LiDAR, and a camera. In the common datasets with semantic labelling, only images from a frontal camera are available, while the employed LiDARs cover a 360° field of view. Another problem is the low resolution of the synthetic intensity images, due to the limited spatial density of LiDAR measurements. As modern convolutional neural networks are designed for high resolution images the use of images of much lower resolution makes problematic segmentation of small objects (in particular cyclists and pedestrians far from the sensors).

Therefore, we have implemented an unsupervised approach that uses the neural network weights learned from typical video sequences only at the pre-learning stage, but then learns directly from the synthesized intensity images that cover the full field of view of the LiDAR. This approach is implemented with the recent open-source Competitive Collaboration architecture [37] (https://github.com/anuragranj/cc, accessed on 16 August 2021). Owing to the Competitive Collaboration framework, which learns jointly motion segmentation, optical flow, and scene depth, we are able to directly use the synthesized intensity images for learning. This, in turn, allows the new version of our detection method to handle objects that are out of the field of view of the frontal camera, and to better adapt to the reduced resolution of the intensity images. However, the most important gain from using the Competitive Collaboration architecture is that we no longer rely on pure semantics, as this neural network uses short sequences of images as inputs, and outputs a soft mask that quantifies per pixel the motion in an image. Hence, the detected objects are considered non-stationary according to their actual dynamics, not just because of belonging to a given class with an assumed ability to move.

### 4.1. Supervised Learning

Supervised cross-modal learning is based on the semantic segmentation network ERFNet. This is an encoder–decoder architecture that is designed to achieve real-time processing speed in autonomous driving applications. The encoder consists of three downsampler blocks and 13 non-bottleneck-1D blocks. Downsampler blocks are placed as the first, second, and eighth layers of the network. A downsampler block is built from a single 3 × 3 convolution layer with stride 2 and a MaxPooling layer. The non-bottleneck-1D block is a distinct feature of ERFNet, that allows it to work in real-time [36]. It is a residual block composed of four 1D convolution layers with filters: 3 × 1, 1 × 3, 3 × 1, 1 × 3. Replacing 2D convolutions with 1D convolutions reduces the number of parameters, hence reducing the processing time. The decoder consists of 3 deconvolution layers with stride 2, separated with two pairs of non-bottleneck-1D blocks.

Supervised cross-modal learning is implemented in two stages. The first stage is pretraining on 10,000 labeled images from KITTI [11], CityScapes [39], and BDD100K [40] datasets (Figure 4). Original images from these datasets are preprocessed by rescaling and converting to grayscale in order to maximally resemble the synthesized intensity images. The semantic class labels provided in the datasets are grouped into three general classes important for the SLAM systems: Class_1: stationary objects (e.g., road, building, wall, traffic sign), Class_2: possibly stationary objects (vegetation, car, truck), and Class_3: non-stationary objects (e.g., motorcycle, bicycle, person). Example images used in this stage are shown in Figure 5A,B. Segmentation into three classes instead of two (stationary and non-stationary) allows us to apply a flexible point rejection strategy in the SLAM system. Depending on the number of available points, uncertain measurements could be removed from the point cloud used by the localization and mapping system.

The second stage of supervised learning is fine-tuning of the model on 150 hand-labeled intensity images synthesized from the publicly available Ouster OS1 sequences. Examples of those images are shown in Figure 5C,D. To prepare labels for training we created a simple tool using the OpenCV library. This tool displays images to be labeled from the provided path. Then, the user selects corners to outline a patch belonging to Class_1 or Class_2 using the left or right mouse button respectively. When the polygon is finished, the user applies it with the keyboard button and can select another one. Unselected fragments are assumed as Class_3.

### 4.2. Unsupervised Learning

The second approach focuses on the detection of non-stationary objects based on motion segmentation. For this task, we leveraged the Competitive Collaboration [37] framework. The learning method is based on the competition of two players supervised by the moderator. The first player consists of two separate neural networks that estimate disparity and camera pose, which requires static fragments of the images. The second one predicts optical flow, which focuses on image fragments in motion. The moderator is a motion segmentation network that splits an image into moving and non-moving areas to feed appropriate data to the two players.

This method uses five consecutive image frames for prediction, and the output mask is generated for the third frame in the sequence. The architecture of the disparity network is inspired by DispNet [41], and the optical flow network is inspired by FlowNetC [42]. The camera motion prediction network is a simple stack of convolutional layers, while the motion segmentation model is an encoder–decoder architecture similar to U-Net. The training procedure consists of two phases. Firstly, the two players optimize their weights, while the moderator has frozen weights. Then, the moderator is trained while the weights of players are frozen.

DispNet is a convolutional network introduced in [41] for the task of disparity estimation. It consists of two parts—contractive and expanding. The contractive branch is built by convolutional layers, and the downsampling is obtained by setting stride to 2. The input resolution is reduced by a factor of 32. The expanding branch is built with the upconvolution layers which upscale spatial resolution of feature maps to the half of original input image size. Those layers are separated by usual convolution layers. On each stage of an upsampling, there is calculated a loss value corresponding to different resolutions. In the first stage of training, only the loss calculated on feature maps with the lowest resolution is considered to learn a coarse representation. At later stages of training, the weights of the losses calculated on higher resolution feature maps are increased to learn fine resolution representation.

FlowNetC is a CNN designed for optical flow estimation [42]. It also has a contractive and expanding part. The contractive part consists of two identical and parallel branches which process two consecutive frames. Each branch is built from three convolutional layers. In the next step, feature maps from each frame are merged using the correlation layer. Later, the merged feature maps from two images are passed by convolutional layers. The expanding part uses upconvolutional layers, and resulting feature maps at each stage are concatenated with appropriate feature maps from the contractive part.

For our purposes, we adopted the existing weight as cross-modal transferred knowledge and then fine-tuned only the moderator—motion segmentation model on intensity images synthesized from the KITTI dataset (Figure 6). To have full coverage of the LiDARs horizontal field of view, we generated six images from a single scan, each covering 60${}^{\circ }$ field of view. For each part, a separate model was trained. For training, we used the sequences 01, 02, 04, 08, 09, and 10 from KITTI, while the sequences 05, 06, and 07 were used for tests, obtaining qualitative and quantitative results.

The output of the motion segmentation network is a continuous grayscale soft mask with values in the 〈0…1〉 range. Similarly like in the previous approach, we can decide how many points should be rejected. In this case, rejection is more flexible because we can set any threshold to output in order to get a binary mask. The difference between unsupervised segmentation of the intensity images generated from the Velodyne LiDAR with 64 scan lines and the Ouster with 128 scan lines can be seen in Figure 7. Many more details can be distinguished in Figure 5C,D, as these images have better resolution, which makes it possible to segment cluttered scenes with multiple objects as pedestrians, cyclists, and cars.

## 5. Application to the Slam System

The target application of our LiDAR data segmentation method is a selection of the points that are useful for SLAM, particularly in urban environments. The SLAM solutions we use in the presented experiments are the LOAM algorithm [8], considered the state-of-the-art in model-based LiDAR odometry, and our PlaneLOAM algorithm, which uses high-level features that group the measured points [5]. With a map consisting of planar patches and line segments, our system improves the accuracy of data association and can optimize the whole map using the factor graph SLAM formulation with the g^2^o library [43]. The overall software architecture of PlaneLOAM that combines real-time scan-to-scan sensor pose tracking and slower, but more accurate scan-to-map localization is the same as in LOAM. However, LOAM does not use all the acquired scan points, focusing on heuristically chosen “planes” and “edges”, which are identified in the point cloud applying simple smoothness criteria [8], that considers the Euclidean distances between neighboring points. While this heuristic rejects some measurements that are potential outliers in ICP matching, it is insufficient in environments with a number of non-stationary objects. Moreover, all the accepted LiDAR points are used with the same weights, while Deschaud [44] demonstrated that keeping too many points that do not provide useful constraints (e.g., because they belong to movable objects or have large measurement errors) can degrade the accuracy of the computed transformation in LiDAR-based SLAM. Hence, in [5] we proposed robust data association methods for the creation and updating of the planar segment and line segment features, which allow PlaneLOAM to collect reliable LiDAR readouts in the form of high-level features. The feature-based map can be optimized, merging similar (multiplied) features and closing the detected loops, but it still may contain invalid features created upon LiDAR points originating from non-stationary objects. Therefore, we demonstrate here that our system can learn a mask that filters out or weights the individual observations with respect to the semantics of the objects they were obtained from. This allows the new version of PlaneLOAM to use only the stable and important observations.

The processing pipeline of PlaneLOAM shown in Figure 8 inherits some blocks from the LOAM algorithm but is entirely different with respect to the creation of the features, creation, and management of the global map and adds a loop closing module based on a modified SegMap method [7,45]. The gains in trajectory and map accuracy due to the new map structure and optimization have been thoroughly evaluated in our recent paper [5], thus they are beyond the scope of this article. However, regarding the presented block scheme of the system (Figure 8), we comment in which blocks elimination of the less reliable laser points can improve the accuracy.

In the application with the PlaneLOAM system, we use the Competitive Collaboration architecture, because it makes it possible to detect non-stationary objects within the full field of view of the LiDAR. The pixel masks that contain information about non-stationary objects are used in the first step of the point cloud processing pipeline. Whenever a new scan is obtained from the LiDAR sensor, six masks covering the 360° field of view are produced by the neural network working in the inference mode. Although continuous grayscale soft masks are produced, we have decided to binarize them using an experimentally chosen threshold $k_{\mathrm{bin}}\in \left(0,1\right)$, which produces binary masks. The default value of $k_{\mathrm{bin}}$ used in our experiments is 0.5. The $k_{\mathrm{bin}}$ threshold makes it possible to decide which points are accepted as belonging to the static environment, and which are rejected as originating from non-stationary object measurements. The six binary masks corresponding to a new scan are transformed into a single image that is the same size as the given laser scan. Then, each point of the scan is transformed according to Equations (Equation 1) and (Equation 2) in order to determine the corresponding pixel position in the binary image. Based on the pixel value, the given point is either excluded from further processing or added to the new point cloud that subsequently is processed by the feature detection and laser odometry blocks.

The block that creates the plane segment and line segment features from the points registered in each scan tries to add these new points to the already existing features (from the same scan) or creates a new feature from the five nearby points. New features are created only if the measurements cannot be associated with any of the existing features, which contributes to a map having a smaller number of larger features. However, if some points in a scan originate from a moving object, then either these points may be assigned to a stationary map feature they actually do not belong to, or an entirely invalid feature can be created, which becomes irrelevant once the object moves away. Also laser points originating from objects having non-rigid surfaces, mainly vegetation, contribute to the creation of features having parameters (i.e., the parametrization of their planar or linear equations) of higher uncertainty. Notice, that we are unable to represent this uncertainty as covariances, as the points measured on non-stationary objects do not necessarily have a bigger spatial dispersion. Thus, these less accurate features can go unnoticed up the processing pipeline, and then at some moment cause wrong data association in the map.

Elimination of the non-stationary points is also important for the scan-to-scan laser odometry and scan-to-map localization blocks. Both procedures compute optimization constraints based upon the point-to-line and point-to-plane distances. In the laser odometry, the procedure is the same as in LOAM, while map-based pose estimation in PlaneLOAM associates selected points from the most recent scan (considered as belonging to either linear or planar features) to the globally consistent map of high-level features. In both cases, points from non-stationary objects can contribute to the creation of invalid constraints, that either associate a non-stationary point from the new scan to a valid feature, or, in the latter case, a non-stationary feature already integrated in the map can generate a number of such invalid constraints.

An important difference between PlaneLOAM and the LOAM system is that while in LOAM the points belonging to linear and planar structures are selected for localization, the global map has a form of a point cloud, in PlaneLOAM we maintain a map of high-level features. This approach promotes better accuracy of the features and allows map optimization, but also requires a mechanism to merge similar features created at different time instances. Therefore, PlaneLOAM implements the merging of co-planar and co-linear features that overlap. In this process, features created from points measured on non-stationary objects can be incidentally merged into valid features. As the parameters of a merged feature are re-estimated from the parameters of the parent features, a feature originating from a non-stationary object that survived in the system to this moment can negatively impact the accuracy of other, valid features.

We do not explicitly use the output of the filter for non-stationary objects in the loop closing, as the SegMap algorithm has its own scan clustering procedure (known as incremental Euclidean segmenter [45]). Therefore, in the experiments with PlaneLOAM, we do not incorporate the constraints from loop closures, as they could obscure the gain in trajectory accuracy obtained using our approach.

## 6. Experimental Results

We evaluate our approach to non-stationary object detection and its integration with LiDAR SLAM on data from two types of LiDARs: the Velodyne HDL-64E, using data from the KITTI dataset, and the Ouster OS1 in the 64 and 128 scan lines versions. The OS1 sequences are those that were made publicly available by Ouster (http://data.ouster.io/sample-data-1.12/index.html, accessed on 16 August 2021), and, unlike the KITTI dataset, they do not have ground truth information for the sensor motion and associated camera images.

Therefore, we used the Ouster data only to produce qualitative results that show how the proposed methods work on the synthesized intensity images and how the results are then used to segment the point clouds. The Velodyne HDL-64E data were also used to obtain qualitative results, which demonstrate the differences between the results depending on the LiDAR type. However, the KITTI dataset is mainly used to obtain quantitative results corroborating our statement that the elimination of LiDAR points originating from non-stationary objects improves the LiDAR SLAM results.

We use PlaneLOAM to quantitatively demonstrate the gains due to the proposed approach to motion segmentation, but we also show that this approach can be easily implemented with the open-source LOAM system, also improving the results. Moreover, we use the open-source LOAM for some qualitative visualizations that make it possible to obtain a better insight into the points removal mechanism.

### 6.1. Qualitative Results in Lidar-Based Slam

In the first experiment, we look more closely at the LOAM point maps obtained from the Ouster OS1-64 sequences. If only the standard LOAM heuristics are applied (Figure 9A,C), then correspondences between points are established on non-stationary objects, for example, on pedestrians, as shown in the inset image in Figure 9A. Then, we apply to the same data the supervised cross-modal learning method with the ERFNet architecture, in order to eliminate the non-stationary points, and thus the invalid associations. Indeed, the segmentation of the intensity images using ERFNet allows the LOAM to reject non-stationary objects (Class_2 and Class_3), removing the vast majority of the invalid correspondences (Figure 9B,D).

The removal of points measured on non-stationary objects works similarly if the unsupervised learning method with the Competitive Collaboration architecture is used. The filter was trained using the KITTI grayscale and synthesized intensity images. Figure 10 depicts how this method handles non-stationary objects in both the intensity images and point clouds. An example of six motion-segmented intensity images synthesized from a Velodyne HDL-64E scan is shown in Figure 10A1–A6. Based on that image, respective points were removed from point clouds, as shown by the red areas in Figure 10C. This results in a more accurate LOAM map representation, as it does not contain points collected from moving objects, such as cars and pedestrians, which are visible in the LOAM points map updated without the rejection procedure (Figure 10B).

Figure 11 demonstrates with sequence 07 from KITTI how the LiDAR measurements originating from non-stationary objects manifest themselves in the map built by our PlaneLOAM algorithm. This map consists of high-level features, hence the points from non-stationary objects are integrated into invalid “phantom objects” that do not represent any persistent part of the environment, and may be wrongly associated with new scans (Figure 11A,B). However, detection of these points by the filter based on the Competitive Collaboration architecture makes it possible to exclude the invalid points at an early stage of the processing pipeline (during the creation of features). As the result, the final map does not contain the invalid features (Figure 11C).

### 6.2. Quantitative Results in Lidar-Based Slam

Knowing the negative influence the points measured on non-stationary objects have on the map created by LiDAR-based SLAM systems, and particularly how these points introduce invalid features in the feature-based map of PlaneLOAM, we examine quantitatively whether our approach makes it possible to counteract these problems at the scale of the entire trajectory, improving the results. For demonstration, we use three sequences from the KITTI dataset [11] (http://www.cvlibs.net/datasets/kitti/eval_odometry.php, accessed on 16 August 2021), which is arguably the most used benchmark for automotive SLAM. We do not compare our SLAM accuracy results to the results obtained by other motion segmentation methods, as we are not aware of any open-source SLAM system working with 3-D LiDAR data that implements explicit rejection of non-stationary objects, and thus can be used for such a comparison.

The ATE (Absolute Trajectory Error) metric proposed in [46] is used for the evaluation of the accuracy of the obtained trajectories. This metric, used also in the PlaneLOAM paper [5], determines how far the estimated sensor/vehicle pose is displaced from its ground truth counterpart stored in the dataset. The error (instantaneous ATE value) is the Euclidean distance between the corresponding points of the considered trajectories. However, to show the quantitative results in a compact form we compute the RMSE (Root Mean Squared Error) of the ATE values along the entire trajectory $\left({\mathrm{ATE}}_{\mathrm{RMS}}\right)$. To complete the comparison we also show the maximum ATE value $\left({\mathrm{ATE}}_{\max}\right)$ for the given sequence.

We have decided to integrate the method based on the Competitive Collaboration architecture as a filter with PlaneLOAM and LOAM, mostly because it handles properly objects within the full field of view of the LiDAR sensor, which made it possible to use the KITTI sequences (not used for evaluation) for training. On the other hand, the grayscale images from the KITTI dataset used for the training of the supervised method with ERFNet covered only the much narrower, forward-looking field of view.

Quantitative results of the application of the filter based on Competitive Collaboration to both the LOAM (open-source version) and PlaneLOAM systems were obtained using sequences no. 05, 06, and 07, which were not used in training. The results are presented in Table 1. From these results, it can be noted that rejecting laser points belonging to non-stationary objects improves the accuracy of the estimated trajectories in both SLAM systems investigated here. The amount of improvement differs between the sequences, as it depends on the number of detected non-stationary objects in the given sequence and on the environment type in which the scans were recorded.

The urban KITTI sequences do not have a large number of non-stationary objects, thus the improvement in terms of ${\mathrm{ATE}}_{\mathrm{RMS}}$ is quite small, however, it is visible for all trajectories (Figure 12). While the improvement due to our feature-based architecture (i.e., the improvement of PlaneLOAM over LOAM without filtering) is in general larger than the improvement due to adding the filter, the filter has a slightly bigger positive impact in the case of PlaneLOAM. This can be explained by the fact that in LOAM the invalid points deteriorate the performance of the ICP estimation only locally, while in PlaneLOAM they can lead to invalid features that are more persistent in the map. Finally, the PlaneLOAM algorithm enhanced by the learnable filter produces very accurate trajectories, as depicted in Figure 13.

One should also notice that the PlaneLOAM results for KITTI sequence 05 are less improved than for the other two sequences in comparison to the open-source LOAM. Although the estimated trajectory is topologically consistent with visually small ATE values (Figure 13A), the errors plot (Figure 12A) reveals that for a part of the trajectory the LOAM system with the filter was more accurate than the basic PlaneLOAM version. This was caused by the environment containing a large number of less-structured objects, including vegetation, which hardly could be represented by planar segments.

Statistics for the management of features in PlaneLOAM without and with the proposed filter are gathered in Table 2. These results suggest that the removal of points mostly influences the number of created planar segments, which represent larger objects. This is consistent with the qualitative results shown in Figure 11. The number of removed LiDAR points is below 10% for all sequences, as the KITTI dataset does not contain scenes with heavy traffic. However, even removing this small fraction of the data deemed unusable in SLAM improves the localization accuracy at a negligible computation cost.

### 6.3. Tests on the Kitti Multi-Object Tracking and Segmentation Dataset

Because the KITTI dataset commonly used for testing SLAM and visual or LiDAR odometry solutions is characterised by rather low dynamics of the scenes and a limited number of non-stationary objects, we extended the tests to the KITTI Multi-Object Tracking and Segmentation (MOTS) dataset (http://www.cvlibs.net/datasets/kitti/eval_mots.php, accessed on 16 August 2021). This part of the KITTI suite is intended for testing and benchmarking algorithms that detect and track moving objects. Because of that, the sequences in MOTS have been collected in more dynamic scenarios, including numerous cars and some cluttered scenes with pedestrians. These sequences contain LiDAR and camera data from the same sensors as the SLAM/odometry dataset, but the ground truth data contains also pixel-wise masks for dynamic objects seen by the frontal camera. Although the KITTI MOTS sequences are shorter than the sequences intended for SLAM evaluation, we picked three examples that have at least 800 frames (LiDAR scans) each, and are representative to different driving scenarios in dynamic environments. The sequences no. 01 and no. 07 represent driving through mostly suburbia areas with the road partially surrounded by vegetation (07) and cars parked on the sides (01). In contrast, sequence 19 was acquired in the downtown area with cars, pedestrians, and isolated examples of vegetation.

Qualitative results from an example suburbia scenario are depicted in Figure 14. One can see that in spite of the low-resolution intensity image synthesized from a Velodyne HDL-64E scan (Figure 14B) the Competitive Collaboration model trained on the grayscale and intensity images from KITTI was able to detect the non-stationary object (a car in motion—Figure 14C) distinguishing it from cars parked at the roadside (Figure 14D) that can still be useful for localization (Figure 14E).

Figure 15 exemplifies the performance of our method in a more complicated downtown environment. As in this scenario non-stationary objects can appear not only along the road, the ability of the Competitive Collaboration model to correctly detect such objects within the 360° field of view becomes more important. This can be seen in Figure 15B,C, where a moving car is detected and segmented out on the synthesized intensity images, although it is only partially visible on the frontal camera image (Figure 15A). These abilities of our system lead to correct removal of the moving objects even in a cluttered scene (Figure 15D), while the LiDAR points belonging to objects that are semantically similar, but stationary, are still feed to the SLAM system (Figure 15E).

The positive effects of this ability to distinguish actual non-stationary objects from their stationary counterparts of the same or similar semantic labelling is demonstrated qualitatively in Figure 16. Here the map built by PlaneLOAM contains planar features related to the parked cars but is free from the invalid features originating in the measurements of the few moving cars (Figure 16B,C).

Finally, we present the quantitative results of PlaneLOAM performance on the three KITTI MOTS sequences measured using the ATE metrics in Table 3. For this experiment we used, alternatively, the pixel-wise motion masks produced by the Competitive Collaboration model (denoted our masks) and the ground truth masks from the KITTI MOTS dataset. The ground truth masks are a result of manual labelling, so they are more accurate, and represent the “upper bound” of the performance in motion detection from 2-D images. However, they are provided only for the frontal camera, thus do not cover the whole field of view of the LiDAR. In contrast, our learned masks refer to the intensity images of lower resolution and have some inaccuracies, particularly on the borders of these images, but cover the entire 360° field of view. One can see in Table 3 that the improvement in the trajectory accuracy with our masks is very similar to the improvement with the ground truth maks, which corroborates the claim that our model generates correct motion segmentation on the intensity images. However, in the case of sequence 19, which has more non-stationary objects all around the sensor, our method outperforms the ground truth masks, as in spite of some inaccuracies it is able to remove more invalid laser points.

### 6.4. Computation Time

Our method needs 96 ms to synthesize a 1024 × 512 [pix] image on an i5 CPU. For training and inference with the Competitive Collaboration and ERFNet neural network models we used Nvidia GTX 1080 Ti GPU. The inference time of the ERFNet model on the tested KITTI sequences for a single intensity image on this GPU was only 20 ms. For the more complicated Competitive Collaboration model, the inference time was 42 ms, while the time needed to project a learned mask onto the point cloud of a LiDAR scan was less than 6 ms. The learning times are reasonable for practical applications—unsupervised training of the Competitive Collaboration model took eight hours in the first phase (cross-modal learning) and then 16 h on a sequence of 5800 synthesized intensity images in the second phase (cf. Figure 6).

## 7. Conclusions

This article demonstrates how LiDAR intensity data, which is often neglected for robot navigation, can be helpful in dynamic environments by allowing the rejection of points that are related to non-stationary objects. This goal is achieved using a simple procedure of synthesizing intensity images from raw LiDAR measurements, and applying cross-modal transfer learning to conserve manual effort and time when preparing the training data. This approach is of practical value, as in spite of the low effort training, our approach can segment the intensity images at the sensor scanning rate, serving as a filter on the input of a SLAM system. The experiments involving the popular LOAM algorithm and our recent PlaneLOAM system, which leverages the map representation with high-level features, provide evidence that the presented approach improves the accuracy of the estimated trajectories in LiDAR-based SLAM. The experiments focused on the unsupervised variant of our method using the Competitive Collaboration. This method offers a practical solution for training a filter in systems using typical LiDAR sensors with 360° field of view. Qualitative results demonstrate that the invalid features created in the PlaneLOAM map from the LiDAR points measured on non-stationary objects are indeed removed by our filter. On the other hand, quantitative results show an improvement in the ATE metrics for all tested sequences. However, this improvement is rather small and, depending on the environment characteristics, we conjecture that it would be beneficial to adjust the LiDAR point removal mechanism, for example, by adaptively setting the $k_{\mathrm{bin}}$ threshold depending on the amount of detected motion. This is one of the directions for our future research.

## Figures and Tables

**Figure 1 sensors-21-06781-f001:**
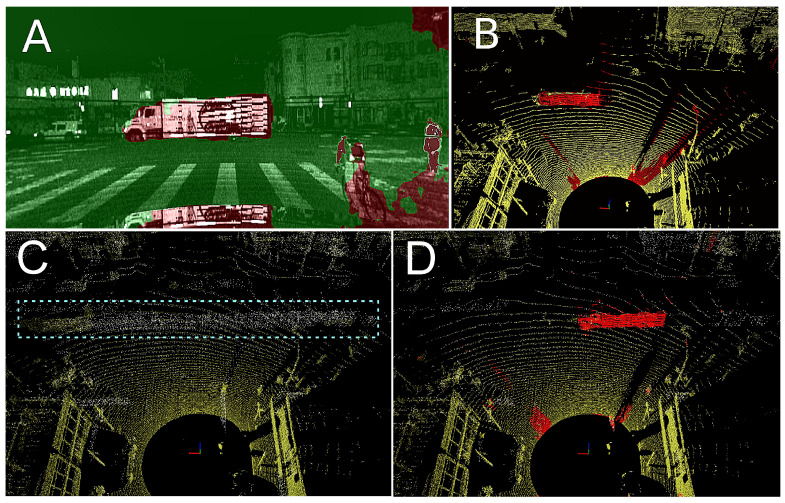
Motivational example of non-stationary objects segmentation and removal on a short sequence from Ouster OS1-128: segmented intensity image (**A**), rejected points (red) in a single scan (**B**), and the resulting map without (**C**) and with (**D**) the proposed method of rejecting non-stationary objects. Note that the slowly moving large truck left a wake of invalid points in the map (blue rectangle), which is not present in the map if the rejection method is used.

**Figure 2 sensors-21-06781-f002:**
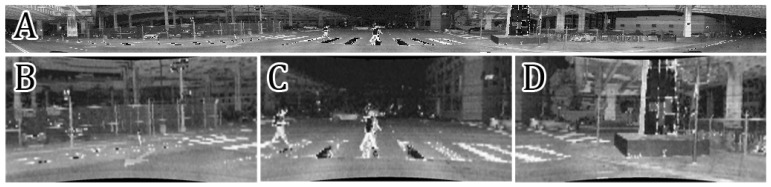
Comparison of a directly generated laser intensity image from Ouster OS1-64 (**A**), and the camera-like images (**B**–**D**) for the same scan from one of the publicly available OS1-64 sequences synthesized with our algorithm.

**Figure 3 sensors-21-06781-f003:**
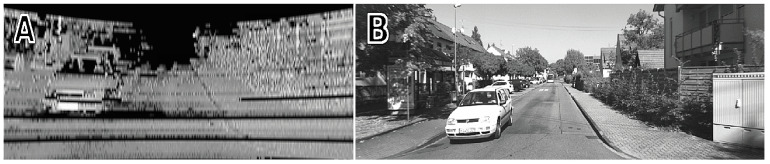
Visual comparison of an image generated from Velodyne HDL-64E intensity data (**A**) and the corresponding grayscale camera image (**B**) from the KITTI dataset.

**Figure 4 sensors-21-06781-f004:**
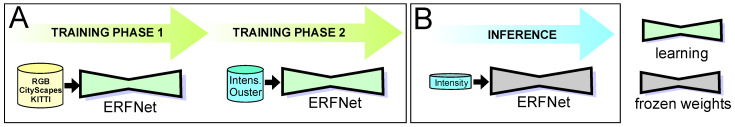
Training of the ERFNet neural network model using the labelled grayscale (Phase 1) and intensity (Phase 2) images (**A**), and the inference process for detection of non-stationary objects using intensity images and the learned semantics (**B**).

**Figure 5 sensors-21-06781-f005:**
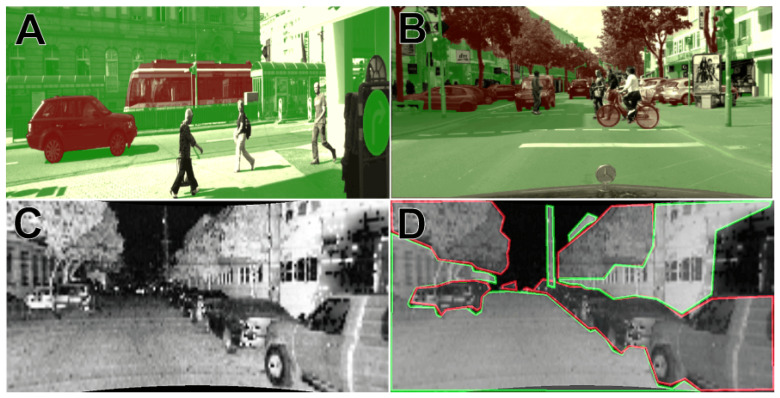
Example from CityScapes (**A**) and KITTI (**B**) used in pre-training, intensity image from Ouster OS1-64 (**C**) with manual labeling (**D**): Class_1 is green, Class_ 2 is red, Class_3 without outline.

**Figure 6 sensors-21-06781-f006:**
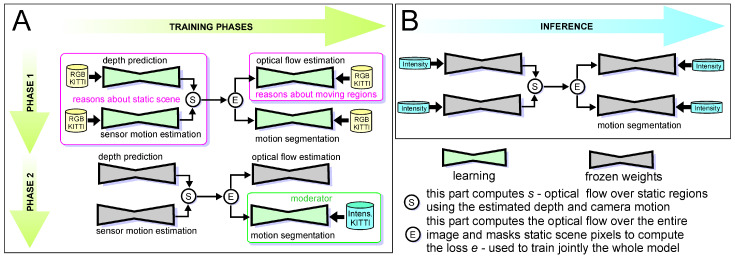
Unsupervised training of the Competitive Collaboration neural network model using the grayscale (Phase 1) and intensity (Phase 2) images (**A**), and the inference process for detection of non-stationary objects using short sequences of intensity images feed to the model (**B**).

**Figure 7 sensors-21-06781-f007:**
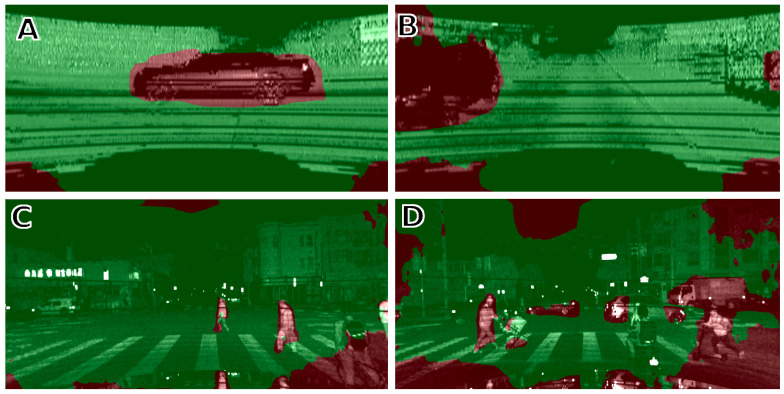
Unsupervised motion segmentation (Competitive Collaboration) from Velodyne HDL-64E (**A**,**B**) and OS1-128 (**C**,**D**). The network correctly detects cars passing in front of the sensor (**A**), but also those moving on the sides, out of the field of view of a typical frontal camera (**B**). OS1-128 images make it also possible to segment out pedestrians (**C**,**D**).

**Figure 8 sensors-21-06781-f008:**
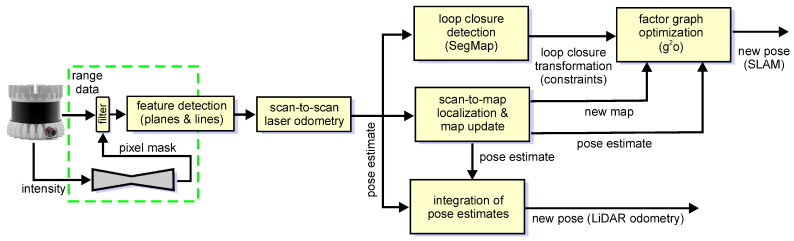
Block scheme of the modified PlaneLOAM architecture with an intensity-based neural filter (indicated by the green dashed line) that removes in real-time measurements originating from non-stationary objects.

**Figure 9 sensors-21-06781-f009:**
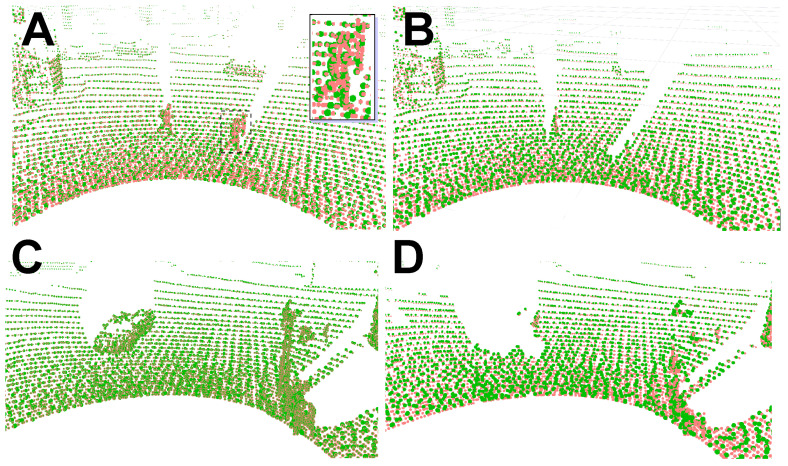
Point clouds before (**A**,**C**) and after (**B**,**D**) rejection of non-stationary objects. Current laser scan in orange, correspondences to the map points in green. Invalid correspondences are clearly visible in the enlarged fragment (inset image).

**Figure 10 sensors-21-06781-f010:**
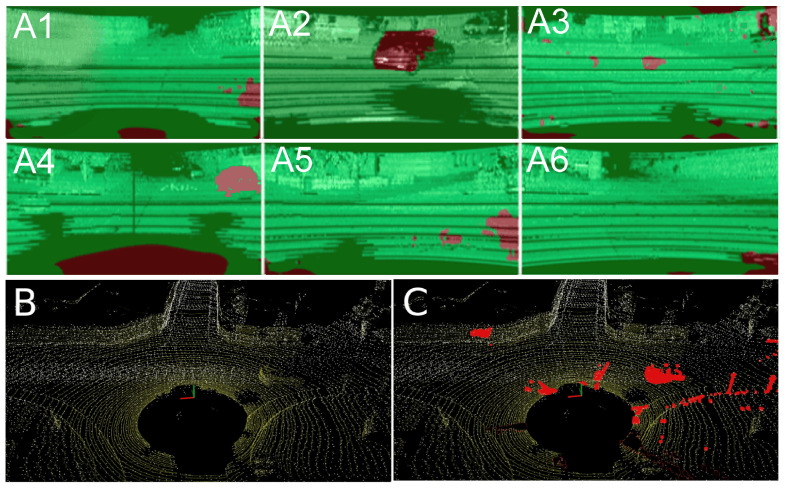
Motion segmentation of a 360° Velodyne HDL-64E scan (**A**). Map (white points) created without (**B**) and with removing of non-stationary objects (**C**). Current laser scan is shown in yellow and rejected points in red.

**Figure 11 sensors-21-06781-f011:**
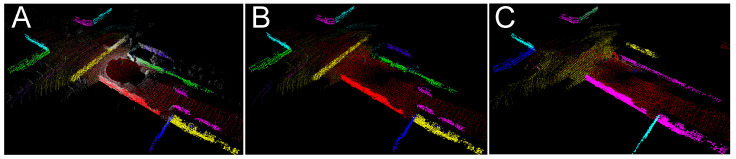
Influence of the motion segmentation filter on the PlaneLOAM map consisting of high-level features indicated by points in random colors: current LiDAR scan shown as white points overlaid on the map with a moving vehicle (yellow) being visible (**A**), a map built without filtration of the non-stationary objects—invalid features created from points measured on cars are visible as yellow and purple segments on the road (**B**), a map built by PlaneLOAM from the filtered scans is free from these features (**C**).

**Figure 12 sensors-21-06781-f012:**
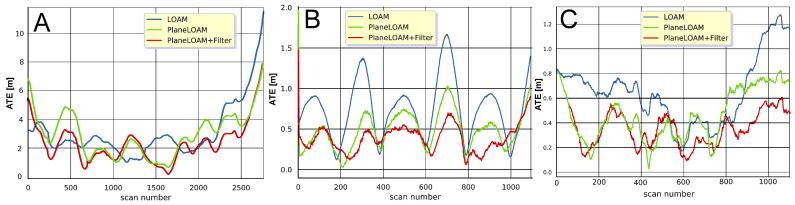
Plots of the ATE values as functions of the scan number for the three KITTI sequences used in evaluations 05 (**A**), 06 (**B**) and 07 (**C**).

**Figure 13 sensors-21-06781-f013:**
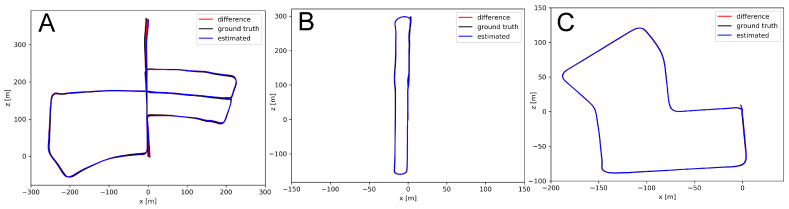
The trajectories estimated by PlaneLOAM with a learned filter for non-stationary objects obtained from the KITTI sequences used in evaluation: 05 (**A**), 06 (**B**) and 07 (**C**). In (**B**,**C**) the red error lines are almost invisible due to the highly accurate trajectory estimation.

**Figure 14 sensors-21-06781-f014:**
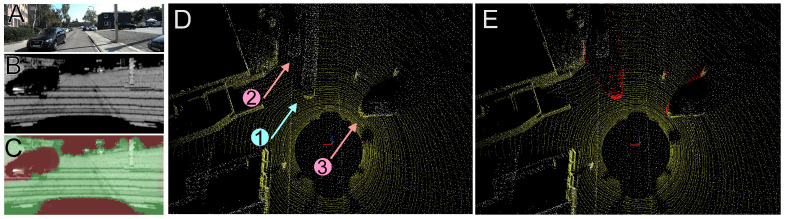
Detection and removal of non-stationary objects in a KITTI MOTS suburban scenario: frontal camera view (**A**), the corresponding synthesized intensity image (**B**), motion segmentation mask (**C**), accumulated point clouds showing the trail of laser points left by a moving car (denoted 1) and some parked cars (denoted 2 and 3) (**D**), and accumulated point clouds after removal of the non-stationary objects—note that the points corresponding to the moving car have been removed, while the parked (stationary) cars are still visible (**E**).

**Figure 15 sensors-21-06781-f015:**
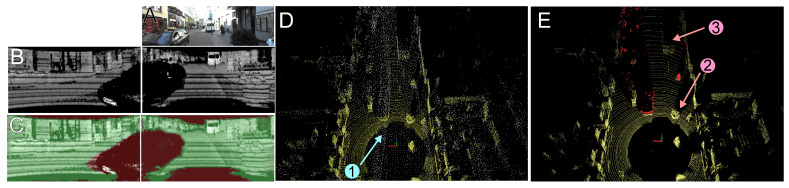
Detection and removal of non-stationary objects in a cluttered environment (KITTI MOTS sequence 19): frontal camera view (**A**), the synthesized intensity images that correspond to this image, but show that the LiDAR has a much larger field of view (**B**), motion segmentation masks for these images (**C**), accumulated point clouds showing the trail of laser points left by a moving car (denoted 1) and invalid points caused by pedestrians (**D**), and a single LiDAR scan clearly showing that while these points have been removed by our method, such stationary objects as a tree (denoted 2) and a parked truck (denoted 3) are not affected (**E**).

**Figure 16 sensors-21-06781-f016:**
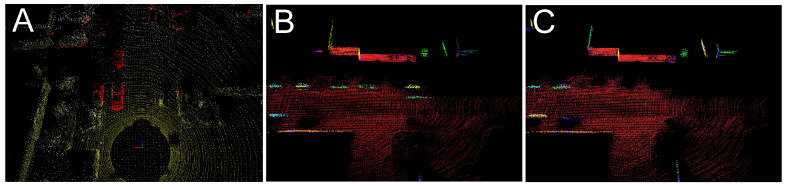
Example views of the PlaneLOAM maps created for the sequence 01 from the KITTI MOTS dataset: removal of non-stationary objects (red points) (**A**), a map created by the PlaneLOAM system without motion filtering showing planar features corresponding to moving cars (**B**), and a map created by the PlaneLOAM + Filter variant that does not contain these features—note that some parked cars are still represented in the map (**C**).

**Table 1 sensors-21-06781-t001:** Comparison of ATE errors for the three KITTI sequences used in evaluation—best results shown in bold.

	LOAM		LOAM + Filter		PlaneLOAM		PlaneLOAM + Filter	
Sequence	${\mathrm{ATE}}_{\mathrm{RMS}}$	${\mathrm{ATE}}_{\max}$	${\mathrm{ATE}}_{\mathrm{RMS}}$	${\mathrm{ATE}}_{\max}$	${\mathrm{ATE}}_{\mathrm{RMS}}$	${\mathrm{ATE}}_{\max}$	${\mathrm{ATE}}_{\mathrm{RMS}}$	${\mathrm{ATE}}_{\max}$
Number	[m]	[m]	[m]	[m]	[m]	[m]	[m]	[m]
05	3.392	11.265	3.010	10.518	3.258	**7.864**	**2.867**	8.104
06	0.829	1.669	0.825	1.671	0.542	2.537	**0.428**	**1.348**
07	0.684	1.278	0.576	1.023	0.502	0.826	**0.418**	**0.657**

**Table 2 sensors-21-06781-t002:** Statistics of the LiDAR points and PlaneLOAM features for three KITTI sequences used in evaluation.

	PlaneLOAM		PlaneLOAM + Filter		
Sequence	Avearage Feat. per Scan		Avearage per Scan		
Number	Lines	Planes	Lines	Planes	Removed Points [%]
05	491	924	483	816	7.2
06	593	891	589	733	8.1
07	335	797	327	709	7.7

**Table 3 sensors-21-06781-t003:** Comparison of ATE errors for trajectories computed by three configurations of PlaneLOAM for the selected sequences from the KITTI MOTS dataset.

	PlaneLOAM (No Filter)		PlaneLOAM + Filter		PlaneLOAM + Filter	
			Ground Truth Masks		Our Masks	
Sequence	${\mathrm{ATE}}_{\mathrm{RMS}}$	${\mathrm{ATE}}_{\max}$	${\mathrm{ATE}}_{\mathrm{RMS}}$	${\mathrm{ATE}}_{\max}$	${\mathrm{ATE}}_{\mathrm{RMS}}$	${\mathrm{ATE}}_{\max}$
Number	[m]	[m]	[m]	[m]	[m]	[m]
01	1.706	2.881	1.597	2.671	1.598	2.880
07	1.979	3.049	1.836	2.916	1.863	2.916
19	1.809	3.981	1.760	3.888	1.618	3.613

## Data Availability

Publicly available datasets were analyzed in this study. This data can be found here: http://data.ouster.io/sample-data-1.12/index.html, http://www.cvlibs.net/datasets/kitti/eval_odometry.php, http://www.cvlibs.net/datasets/kitti/eval_mots.php.

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
