# Peer review of "Real-Time Detection of Non-Stationary Objects Using Intensity Data in Automotive LiDAR SLAM"

_sensors, 2021, doi:10.3390/s21206781_

Round 1
Reviewer 1 Report
This article aims at demonstrating the feasibility of modern deep learning techniques for real-time detection of non-stationary objects in point clouds obtained from 3-D light detecting and ranging (LiDAR) sensors. The proposed method has some novelty, but the following issues should be considered.
- What is the relationship between the standard of non-stationary objects segmentation and the motion state of the target?
- What is the difference between the processing strategy of fast target and slow target?
- The proposed method should be compared with the existing methods to better reflect its advantages.
- The calculation amount of the proposed method shall be given.
- The relationship between the proposed method and signal-to-noise ratio should be analyze.
Author Response
This article aims at demonstrating the feasibility of modern deep learning techniques for real-time detection of non-stationary objects in point clouds obtained from 3-D light detecting and ranging (LiDAR) sensors. The proposed method has some novelty, but the following issues should be considered.
Reply: Thank you for the review and a positive opinion on the novelty of our work. Below we address the issues raised in the review.
2. What is the relationship between the standard of non-stationary objects segmentation and the motion state of the target?
Reply: As stated in the Introduction, our method does not track the targets. The very idea of this work is that we detect the non-stationary objects instantianously in the intensity images and filter their points out for the SLAM algorithm. The new subsection "6.3 Tests on the KITTI Multi-Object Tracking and Segmentation dataset" added to the revised manuscript demosntrates that our method distinguishes the moving objects from the stationary ones, even if they have the same semantic labels (e.g. both are cars).
2. What is the difference between the processing strategy of fast target and slow target?
Reply: There is not specific strategy for different target types, as we do not explicit track the target, but only detect them and remove the laser points belonging to the objects considered non-stationary. However, in the new subsection "6.3 Tests on the KITTI Multi-Object Tracking and Segmentation dataset" added to the revised manuscript we demosntrate that our method distinguish the moving objects from the stationary ones, even if they have the same semantic labels (e.g. both are cars). Also the pedestrians (moving slowly) are correctly detected (see the new Fig. 15). Unfortunately, the publicly available datsets we used do not contain such extreme cases as very slow moving or very fast vehicles.
3. The proposed method should be compared with the existing methods to better reflect its advantages.
Reply: We agree with the reviewer that such a comparison would be beneficial. Unfortunately, we are not aware of any LiDAR-based 3-D SLAM that has specific detection and removal of moving targets. While such solutions exist for visual SLAM (e.g. DynaSLAM II) and RGB-D SLAM, they cannot be directly compared to our method. Also a comparison to objects tracking is not conclusive, as our method does not track the detected objects. Therefore, we have decided to extend the manuscript by testing our method on the KITTI Multi-Object Tracking and Segmentation dataset. These tests indirectly demonstrate that the motion segmentation masks computed by our neural models are of at least comparable quality to the ground truth masks provided by this dataset.
4. The calculation amount of the proposed method shall be given.
Reply: A new subsection "6.4 Computation time" was added in the revised manuscript that gives detailed information of the time required to complete the key operations in our method for both variants (i.e. ERFNet and Competitive Collaboration models).
5. The relationship between the proposed method and signal-to-noise ratio should be analyze.
Reply: Unfortunately, we cannot measure directly the amount of noise in the used images, as we have only access to the datasets, not to raw LiDAR data. However, we conjecture that the spatial resolution of the intensity images has a bigger impact on the performance than the signal to noise ratio, which is corroborated by the results (see Fig. 7).
We also attach a document showing the differences between the original and revised manuscript.
Reviewer 2 Report
The paper proposed a method for real-time detection of non-stationary objects in point clouds obtained from 3-D LiDAR sensors. The subject of this paper fits the scope of journal. Consider to refine and summarize the novelty the novelty of the paper in Introduction and Conclusions.
Author Response
The paper proposed a method for real-time detection of non-stationary objects in point clouds obtained from 3-D LiDAR sensors. The subject of this paper fits the scope of journal. Consider to refine and summarize the novelty the novelty of the paper in Introduction and Conclusions.
Reply: Thank you for the review. In the revised manuscript we substantially modified the Introduction, giving the contribution (novelty) as a bulleted list. We also clarified the backgorund and motivation for the specific approach we propose.
We also attach a document showing the differences between the original and revised manuscript.

Reviewer 3 Report
The article aims at demonstrating the feasibility of modern deep learning techniques for real-time detection of non-stationary objects in point clouds obtained from 3-D light detecting and ranging (LiDAR) sensors. The proposed approach exploits images synthesized from the reflectance (received intensity) data yielded by the modern LiDARs along the usual range measurements. We demonstrate that non-stationary objects can be detected using neural network models, applying 2-D grayscale images in the training process, or using unsupervised learning.
The manuscript is well written and easy to follow. The introduction includes all the necessary references. The conclusions are strong and well described.
The proposed method is relevant to the field and the journal.
The datasets (both images and point clouds) are needed in the annexes section.
Figures 9 to 11 are well detailed, however more scenarios are needed to validate the method and generate more results.
Some aditional relevant references need to be included such as: https://www.mdpi.com/2073-4395/11/1/11
Some "typographical errors" need to be corrected.
Author Response
The article aims at demonstrating the feasibility of modern deep learning techniques for real-time detection of non-stationary objects in point clouds obtained from 3-D light detecting and ranging (LiDAR) sensors. The proposed approach exploits images synthesized from the reflectance (received intensity) data yielded by the modern LiDARs along the usual range measurements. We demonstrate that non-stationary objects can be detected using neural network models, applying 2-D grayscale images in the training process, or using unsupervised learning.
The manuscript is well written and easy to follow. The introduction includes all the necessary references. The conclusions are strong and well described.
Reply: Thank you for the insightful review and positive comments on our work.
The proposed method is relevant to the field and the journal.
Reply: Thank you, we hope this paper would be interesting to the Sensors audience.
The datasets (both images and point clouds) are needed in the annexes section.
Reply: The revised manuscript contains URLs (links in footnotes) to all the datasets we used to produce the experimental results, and links to the code and neural models of the proposed system on GitHub. We hope this makes it possible to replicate our results and develop further the ideas.
Figures 9 to 11 are well detailed, however more scenarios are needed to validate the method and generate more results.
Reply: Indeed, we agree that more challenging scenarios were necessary to demonstrate the strengths of our method. This is implemented in the revised manuscript that contains an entirely new subsection "6.3 Tests on the KITTI Multi-Object Tracking and Segmentation dataset". Three new qualitative examples of diversified scenarios are demonstarted in suburban and downtown environment. Moreover, a new table (Tab. 3) with quantitative results on the KITTI MOTS dataset is provided.
Some aditional relevant references need to be included such as: https://www.mdpi.com/2073-4395/11/1/11
Although in our opinion the pointed paper from MDPI Afronomy is not particularly relevant to our work because of a very different use case (precision agriculture vs. autonomous driving), we went through thelatest results available in the literature and added seven new references to the revised manuscript that more accurately shown the state-of-the-art relevant to our research.
Some "typographical errors" need to be corrected.
Reply: We are sorry for any typos ang grammar errors in the original manuscript. We believe thy have been removed through proofreading and checking with the Grammarly software.
We also attach a document showing the differences between the original and revised manuscript.

Reviewer 4 Report
Overall the paper seems to be interesting and sound, however, in my opinion, it is still affected by some problems.
First of all, in some parts, the clarity and editorial quality of the paper weaken. As a consequence, such parts result to be quite difficult to read. Therefore, I would suggest to carefully improve the prose of writing in order to make this paper easier to read.
Furthermore, presentation aside, by reading the paper, it still was not entirely clear what to expect with the direction of the article. Indeed, the contribution proposed in this paper has been only marginally compared and contextualised with respect to the state of the art. As a result, it is extremely difficult to understand the novelty/contributions introduced by the paper. The aforementioned aspects should be carefully addressed before the paper can be considered any further. The figures should be better explained in their component parts. The use of mathematical notation should always be supported by an appropriate informal description. By doing this, the paper may be easier to read and follow.
Finally, a thorough proofreading would be suggested, since in the paper there are some typos and formatting issues.
As remarks:
- The paper should be better compared and contextualized with respect to the state of the art.
- In some parts of the paper, the clarity and editorial quality of the paper weaken. As a consequence, such parts result to be quite difficult to read. Therefore, I would suggest to carefully improve the prose of writing in order to make this paper easier to read.
- Each figure should be properly defined within the text and must be improved in quality.
- An accurate proofreading is strongly recommended.
Author Response
Overall the paper seems to be interesting and sound, however, in my opinion, it is still affected by some problems.
Reply: Thank you for the review and a positive overall opinion on the paper.
First of all, in some parts, the clarity and editorial quality of the paper weaken. As a consequence, such parts result to be quite difficult to read. Therefore, I would suggest to carefully improve the prose of writing in order to make this paper easier to read.
Reply: Indeed, as the paper was initially a short Workshop contribution, then extended to a full paper it suffered from some weakness in its narration. Therefore, we modified the manuscript substantially, trying to make it more clear and removing any repetitive statements.
Furthermore, presentation aside, by reading the paper, it still was not entirely clear what to expect with the direction of the article. Indeed, the contribution proposed in this paper has been only marginally compared and contextualised with respect to the state of the art. As a result, it is extremely difficult to understand the novelty/contributions introduced by the paper. The aforementioned aspects should be carefully addressed before the paper can be considered any further. The figures should be better explained in their component parts. The use of mathematical notation should always be supported by an appropriate informal description. By doing this, the paper may be easier to read and follow.
Reply: We agree that more experimental results were necessary in the paper. Hence, in the revised manuscript we added two entirely subsections 6.3 and 6.4, adressing the performance on the KITTI Multi-Object Tracking and Segmentation dataset (MOTS) and the computing time, respectively.
Finally, a thorough proofreading would be suggested, since in the paper there are some typos and formatting issues.
Reply: The revised manuscript was checked for typos and grammar errors by proofreading and using the Grammarly software.
As remarks:
- The paper should be better compared and contextualized with respect to the state of the art.
Reply: In the revised manuscript we added two entirely subsections 6.3 and 6.4, adressing the performance on the KITTI Multi-Object Tracking and Segmentation dataset (MOTS) and the computing time, respectively. Unfortunately, we are not aware of any open-source LiDAR 3-D SLAM system that has explicit removal of non-stationary objects to directly compare it to our solution. However, we added links to our open-source code on GitHub and links to all datsets which facilitates such a comparison in the future.
- In some parts of the paper, the clarity and editorial quality of the paper weaken. As a consequence, such parts result to be quite difficult to read. Therefore, I would suggest to carefully improve the prose of writing in order to make this paper easier to read.
Reply: The mansucript has been substantially revised improving the clarity of presentation.
- Each figure should be properly defined within the text and must be improved in quality.
Reply: The figure captions have been reviewed and extended when necessary to make them self-containing. Moreover, the figures showing the neural network architectures (Fig. 5 and Fig. 6) have been updated to improve the clarity.
- An accurate proofreading is strongly recommended.
Reply: Proofreading by human beings and software was implemented.
We also attach a document showing the differences between the original and revised manuscript.

Round 2
Reviewer 3 Report
The authors have taken into consideration the suggestions made, the work has been considerably improved compared to its previous version.